# Accuracy of real-time multi-model ensemble forecasts for seasonal influenza in the U.S.

Nicholas G. Reich[1]*, Craig J. McGowan[2], Teresa K. Yamana[3], Abhinav Tushar[4], Evan L. Ray[5], Dave Osthus[6], Sasikiran Kandula[3], Logan C. Brooks[7], Willow Crawford-Crudell[8], Graham Casey Gibson[1], Evan Moore[1], Rebecca Silva[9], Matthew Biggerstaff[2], Michael A. Johansson[10], Roni Rosenfeld[11], Jeffrey Shaman[3]

**1** Department of Biostatistics and Epidemiology, University of Massachusetts-Amherst, Amherst, Massachusetts, United States of America, **2** Influenza Division, Centers for Disease Control and Prevention, Atlanta, Georgia, United States of America, **3** Department of Environmental Health Sciences, Columbia University, New York, New York, United States of America, **4** School of Computer Science, University of Massachusetts-Amherst, Amherst, Massachusetts, United States of America, **5** Department of Mathematics and Statistics, Mount Holyoke College, South Hadley, Massachusetts, United States of America, **6** Statistical Sciences Group, Los Alamos National Laboratory, Los Alamos, New Mexico, United States of America, **7** Computer Science Department, Carnegie Mellon University, Pittsburgh, Pennsylvania, United States of America, **8** Department of Mathematics and Statistics, Smith College, Northampton, Massachusetts, United States of America, **9** Department of Mathematics and Statistics, Amherst College, Amherst, Massachusetts, United States of America, **10** Division of Vector-Borne Diseases, Centers for Disease Control and Prevention, San Juan, Puerto Rico, United States of America, **11** Machine Learning Department, Carnegie Mellon University, Pittsburgh, Pennsylvania, United States of America

* nick@umass.edu

**Data Availability Statement:** All data from this project are available on GitHub (https://github.com/FluSightNetwork/cdc-flusight-ensemble), with a

## Abstract

Seasonal influenza results in substantial annual morbidity and mortality in the United States and worldwide. Accurate forecasts of key features of influenza epidemics, such as the timing and severity of the peak incidence in a given season, can inform public health response to outbreaks. As part of ongoing efforts to incorporate data and advanced analytical methods into public health decision-making, the United States Centers for Disease Control and Prevention (CDC) has organized seasonal influenza forecasting challenges since the 2013/2014 season. In the 2017/2018 season, 22 teams participated. A subset of four teams created a research consortium called the FluSight Network in early 2017. During the 2017/2018 season they worked together to produce a collaborative multi-model ensemble that combined 21 separate component models into a single model using a machine learning technique called stacking. This approach creates a weighted average of predictive densities where the weight for each component is determined by maximizing overall ensemble accuracy over past seasons. In the 2017/2018 influenza season, one of the largest seasonal outbreaks in the last 15 years, this multi-model ensemble performed better on average than all individual component models and placed second overall in the CDC challenge. It also outperformed the baseline multi-model ensemble created by the CDC that took a simple average of all models submitted to the forecasting challenge. This project shows that collaborative efforts between research teams to develop ensemble forecasting approaches can bring measurable improvements in forecast accuracy and important reductions in the variability of performance from year to year. Efforts such as this, that emphasize real-time testing and evaluation of forecasting models and facilitate the close collaboration between

permanent repository stored on Zenodo (https://doi.org/10.5281/zenodo.1255023).

**Funding:** NGR, AT, GCG, and EM were funded by National Institutes of Health (grant number R35GM119582). NGR, AT, and ELR were funded by the Defense Advanced Research Projects Agency (YFA16 D16AP00144). LB and RR were funded by grants from the Defense Threat Reduction Agency (Contract No. HDTRA1-18-C-0008) and the National Institutes of Health (grant number 5U54GM088491). LB was funded by the National Science Foundation (grant numbers 0946825, DGE-1252522, and DGE- 1745016), and a gift from Uptake Technologies. TKY, SK, and JS were funded by National Institutes of Health (grant number GM110748). The funders had no role in study design, data collection and analysis, decision to publish, or preparation of the manuscript.

**Competing interests:** I have read the journal's policy and the authors of this manuscript have the following competing interests: JS and Columbia University disclose partial ownership of SK Analytics.

public health officials and modeling researchers, are essential to improving our understanding of how best to use forecasts to improve public health response to seasonal and emerging epidemic threats.

## Author summary

Seasonal influenza outbreaks cause millions of infections and tens of thousands of deaths in the United States each year. Forecasting the track of an influenza season can help public health officials, business leaders, and the general public decide how to respond to an ongoing or emerging outbreak. Our team assembled over 20 unique forecasting models for seasonal influenza and combined them together into a single "ensemble" model. We made predictions of the 2017/2018 influenza season, each week sending real-time forecasts to the US Centers for Disease Control and Prevention (CDC). In the 2017/2018 influenza season, one of the largest seasonal outbreaks in the last 15 years, our ensemble model performed better on average than all individual forecast models in the ensemble. Based on results from this study, the CDC used forecasts from our ensemble model in public communication and internal reports in the subsequent 2018/2019 influenza season.

## Introduction

Seasonal influenza results in a substantial annual public health burden in the United States and worldwide. The United States Centers for Disease Control and Prevention (CDC) estimates there were 48.8 million cases of influenza, 959,000 influenza-related hospitalizations, and nearly 80,000 influenza-related deaths in the U.S. from October 2017 through May 2018, making the 2017/2018 season one of the largest on record [1]. The CDC utilizes a variety of surveillance methods to assess the severity of an influenza season, including monitoring outpatient visits for influenza-like illness (ILI), influenza-related hospitalizations, and virologic testing [2]. However, like all surveillance systems, these records describe only a sample of events that have already taken place, and provide limited indication of the future timing or severity of the epidemic, which can vary substantially from season to season [3]. Forecasts of an influenza season offer the possibility of providing actionable information on future influenza activity that can be used to improve public health response. Recent years have seen a substantial increase of peer-reviewed research on predicting seasonal influenza [4–11].

Multi-model ensembles, i.e. models that combine predictions from multiple different component models, have long been seen as having both theoretical and practical advantages over any single model [12–15]. First, it allows for a single forecast to incorporate signals from different data sources and models that may highlight different features of a system. Second, combining signals from models with different biases may allow those biases to offset and result in an ensemble that is more accurate and has lower variance than the individual ensemble components. Weather and climate models have utilized multi-model ensemble systems for these very purposes [16–19], and recent work has extended ensemble forecasting to infectious diseases, including influenza, dengue fever, lymphatic filariasis, and Ebola hemorrhagic fever [20–23]. Throughout the manuscript, we will use the term ensemble to refer generally to these multi-model ensemble approaches.

Since the 2013/2014 influenza season, the CDC has run an annual prospective influenza forecasting competition, known as the FluSight challenge, in collaboration with outside

researchers. The challenges have provided a venue for close interaction and collaboration between government public health officials and academic and private-sector researchers. Among other government-sponsored infectious disease forecasting competitions in recent years, [24, 25] this challenge has been unique in its prospective orientation over multiple outbreak seasons. Each week from early November through mid-May, participating teams submit probabalistic forecasts for various influenza-related targets of interest. During the 2015/2016 and 2016/2017 FluSight challenges, analysts at the CDC built a simple ensemble model by taking an unweighted average of all submitted models. This model has been one of the top performing models each season [26].

The FluSight challenge has been designed and retooled over the years with an eye towards maximizing the public health utility and integration of forecasts with real-time public health decision making. All forecast targets are derived from the weighted percentage of outpatient visits for influenza-like illness (wILI) collected through the U.S. Outpatient Influenza-like Illness Surveillance Network (ILINet), weighted by state populations (Fig 1B and 1C). ILI is one of the most frequently used indicators of influenza activity in epidemiological surveillance. Weekly submissions to the FluSight challenge contain probabilistic and point forecasts for seven targets in each of 11 regions in the U.S. (national-level plus the 10 Health and Human Services (HHS) regions, Fig 1A). There are two classes of targets: "week-ahead" and "seasonal". "Week ahead" targets refer to four short-term weekly targets (ILI percentages 1, 2, 3 and 4 weeks in the future) that are different for each week of the season. "Seasonal" targets refer to quantities (outbreak onset week, outbreak peak week, and outbreak peak intensity) that represent a single outcome observed for a region in a season (see Methods).

In March 2017, influenza forecasters who had worked with CDC in the past were invited to join in establishing the FluSight Network. This research consortium worked collaboratively throughout 2017 and 2018 to build and implement a real-time multi-model ensemble with performance-based model weights. A central goal of the FluSight Network was to demonstrate the benefit of performance-based weights in a real-time, multi-team ensemble setting by outperforming the "simple average" ensemble that CDC uses to inform decision making and situational awareness during the annual influenza season. The CDC used this project to evaluate in real-time the feasibility and accuracy of creating an ensemble forecast based on past performance. Based on the forecast accuracy shown in this experiment, the CDC decided to adopt the approach described here as its main forecasting approach for the 2018/2019 influenza season.

In this paper, we describe the development of this collaborative multi-model ensemble and present forecasting results from seven retrospective seasons and one prospective season. The FluSight Network assembled 21 component forecasting systems to build multi-model ensembles for seasonal influenza outbreaks (Table A in S1 Text). These components encompassed a variety of different modeling philosophies, including Bayesian hierarchical models, mechanistic models of infectious disease transmission, statistical learning methodologies, and classical statistical models for time-series data. We show that using multi-model ensembles informed by past performance consistently improved forecast accuracy over using any single model and over multi-model ensembles that do not take past performance into account. Given the timing of this experiment, during a particularly severe influenza season, this work also provides the first evidence from a real-time forecasting study that performance-based weights can improve ensemble forecast accuracy during a high severity infectious disease outbreak. This research is an important example of collaboration between government and academic public health experts, setting a precedent and prototype for real-time collaboration in future outbreaks, such as a global influenza pandemic.

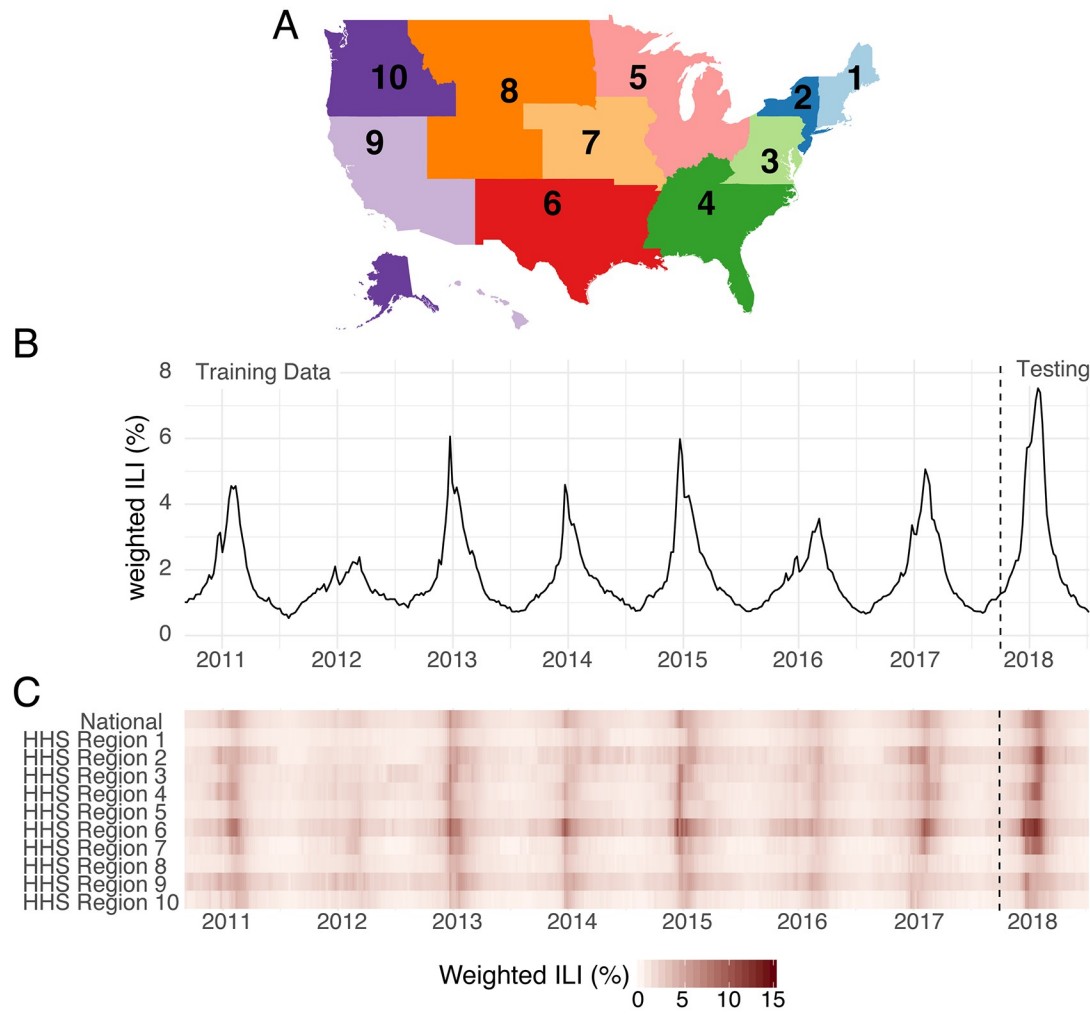

**Fig 1. Overview of region-level influenza surveillance data in the US.** (A) Map of the 10 U.S. Health and Human Services regions. Influenza forecasts are made at this geographic scale. (B) Publicly available wILI data from the CDC website for the national level. The y-axis shows the estimated percentage of doctor's office visits in which a patient presents with influenza-like illness for each week from September 2010 through July 2018. The dashed vertical line indicates the separation of the data used by the models presented here for the training (retrospective) and testing (prospective) phases of analysis. (C) Publicly available wILI data for National level and each of the 10 HHS regions. Darker colors indicate higher wILI.

## Results

### Summary of ensemble components

Twenty-one individual component models were fit to historical data and used to make pro-spective forecasts during seven training seasons (2010/2011–2016/2017, Table A in S1 Text). Their forecast accuracy varied widely across region, season, and target. A detailed comparative analysis of component model forecast performance can be found elsewhere [27]; however, here we summarize a few key insights. A seasonal baseline model, whose forecasts for a partic-ular target are based on data from previous seasons and do not update based on data from the current season, was used as a reference point for all component models. Over 50% of the indi-vidual component models out-performed the seasonal baseline model in forecasting 1-, 2-, and 3-week ahead incidence as well as season peak percentage and season peak week. How-ever, season-to-season variability in relative forecast performance was large. For example, 10

component models had, in at least one season, better overall accuracy than the model with the best average performance across all seasons. To evaluate model accuracy, we followed CDC convention and used a metric that takes the geometric average of the probabilities assigned to a small range around the eventually observed values. This measure, which we refer to as "forecast score", can be interpreted as the average probability a given forecast model assigned to the values deemed by the CDC to be accurate (see Methods). As such, higher values, on a scale of 0 to 1, indicate more accurate models.

## Choice of ensemble model based on cross-validation

We pre-specified five candidate ensemble approaches prior to any systematic evaluation of ensemble component performance in previous seasons and prior to the 2017/2018 season (Table A in S1 Text) [28]. The pre-specified ensemble approaches all relied on taking weighted averages of the component models, including two seasonal baseline components, using a predictive density stacking approach (see Methods). They ranged in complexity from simple (every component is assigned a single weight) to more complex (components have different weights depending on the target and region being forecasted, see Methods). The FSNetwork Target-Type Weights (`FSNetwork-TTW`) ensemble model, a medium-complexity approach, outperformed all other multi-model ensembles and components in the training phase by a slim margin (Fig 2). The `FSNetwork-TTW` model built weighted model averages using 40 estimated weights, one for each model and target-type (week-ahead and seasonal) combination (Fig 3). In the training period, consisting of the seven influenza seasons prior to 2017/2018, this model achieved a leave-one-season-out cross-validated average forecast score of 0.406, compared with the FSNetwork Target Weights (`FSNetwork-TW`) model with a score of

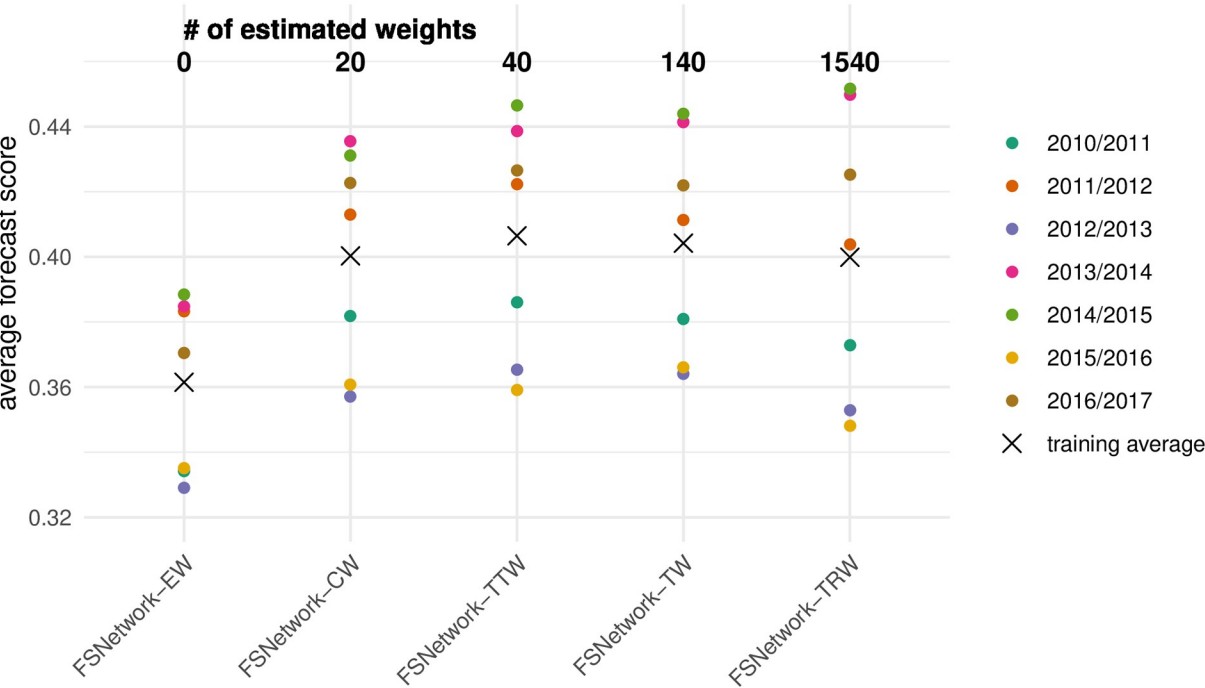

**Fig 2. Training phase performance of the five pre-specified multi-model ensembles.** The five ensembles tested were Equal Weights (EW), Constant Weights (CW), Target-Type Weights (TTW), Target Weights (TW), and Target-Region Weights (TRW). The models are sorted from simplest (left) to most complex (right), with the number of estimated weights (see Methods) for each model shown at the top. Each point represents the average forecast score for a particular season, with overall average across all seasons shown by the X.

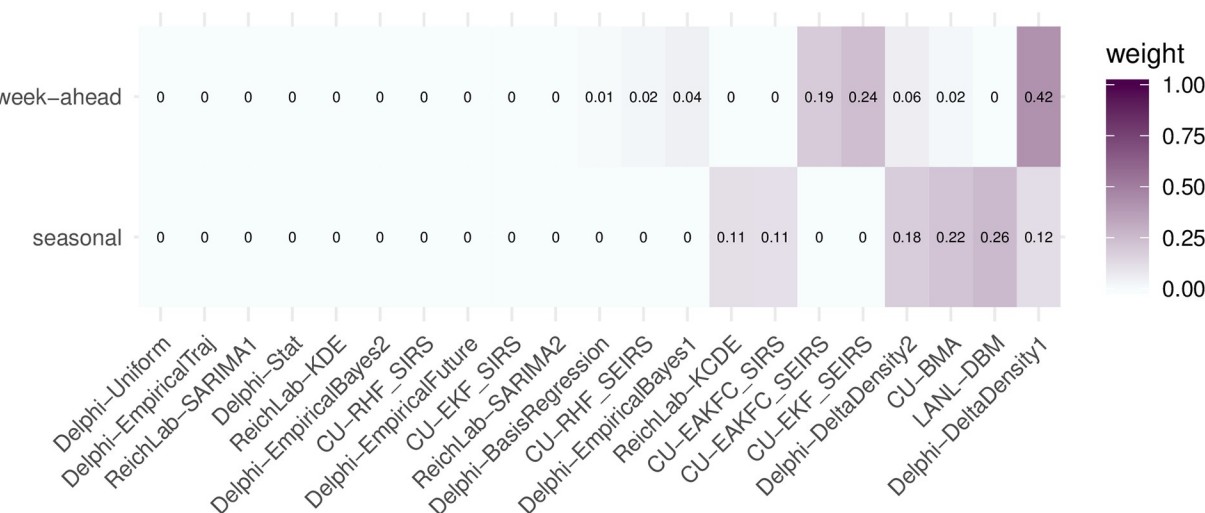

**Fig 3. Component model weights for the FluSight Network Target-Type Weights (`FSNetwork-TTW`) ensemble model in the 2017/2018 season.** Weights were estimated using cross-validated forecast performance in the 2010/2011 through the 2016/2017 seasons.

0.404, the FSNetwork Constant Weights (`FSNetwork-CW`) model with a score of 0.400, and the FSNetwork Target-Region Weights (`FSNetwork-TRW`) model with a score of 0.400 (Fig 4). Prior to the start of the 2017-18 FluSight Challenge, we chose the target-type weights model as the model that would be submitted in real-time to the CDC during the 2017/2018 season. This choice was based on the pre-specified criteria of the chosen model having the highest score of any approach in the cross-validated training phase [28].

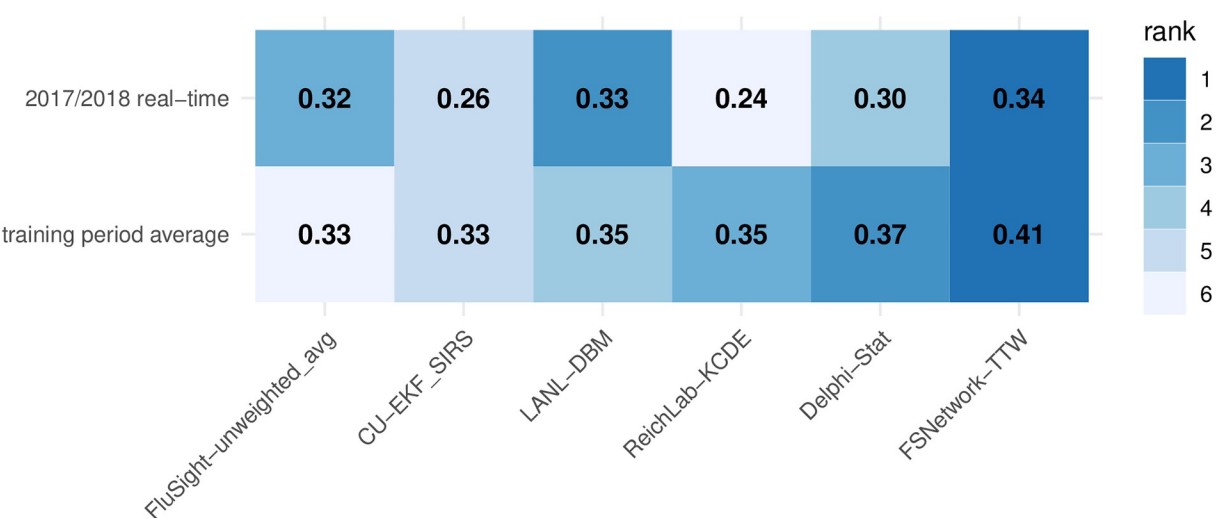

**Fig 4. Overall test and training phase performance scores for selected models.** Displayed scores are averaged across targets, regions, and weeks, and plotted separately for selected models. Models shown include the `FSNetwork-TTW` model, the top performing model from each team during the training phase and, for the last two training seasons and the test season, the unweighted average of all FluSight models received by CDC. Model ranks within each row are indicated by color of each cell (darker colors indicates higher rank and more accurate forecasts) and the forecast score (rounded to two decimal places) is printed in each cell. Note that a component's standalone accuracy does not necessarily correlate to its contribution to the overall ensemble accuracy. See discussion in the Ensemble Components subsection of the Methods.

Using the results from the training period, we estimated weights for the chosen `FSNetwork-TTW` ensemble model that would be used for the 2017/2018 real-time forecasting. The `FSNetwork-TTW` model assigned non-negligible weight (greater than 0.001) to 8 models for week-ahead targets and 6 models for seasonal targets (Fig 3). For week-ahead targets, the highest non-zero weight (0.42) was given to the `Delphi-DeltaDensity1` model. For seasonal targets, the highest weight (0.26) was given to the `LANL-DBM` model. In the weights for the seasonal targets, six models shared over 99.9% of the weight, with none of the six having less than 0.11 weight. All four research teams had at least one model with non-negligible weight in the chosen model.

## Summary of ensemble real-time performance in 2017/2018 season

The 2017/2018 influenza season in the U.S. exhibited features that were unlike that of any season in the past 15 years (Fig 1B and 1C). As measured by wILI percentages at the national level, the 2017/2018 season was on par with the other two highest peaks on record since 1997: the 2003/2004 season and the 2009 H1N1 pandemic. In some regions, for example HHS Region 2 (New York and New Jersey) and HHS Region 4 (southeastern states), the peak wILI for the 2017/2018 season was more than 20% higher than previously observed peaks. Because all forecasting models rely, to some extent, on future trends mimicking observed patterns in the past, the anomalous dynamics in 2017/2018 posed a challenging "test season" for all models, including the new ensembles.

In spite of these unusual dynamics, the chosen `FSNetwork-TTW` ensemble showed the best performance among all component and ensemble models during the 2017/2018 season. In particular, we selected the single best component model from each team in the training phase stage and the `FluSight-unweighted_avg` model (the unweighted average of all models submitted to the CDC) to compare with the `FSNetwork-TTW` model (Fig 4). The results from 2017/2018 were consistent with and confirmed conclusions drawn from the training period, where the `FSNetwork-TTW` model outperformed all other ensemble models and components. The `FSNetwork-TTW` model had the highest average score in the training period (0.406) as well as the highest average score in the 2017/2018 test season (0.337). This strong and consistent performance by the chosen `FSNetwork-TTW` ensemble model is noteworthy given that our team identified this model prospectively, before the season began, essentially wagering that this ensemble model would have the best performance of any model in the 2017/2018 season, which it did. The `FSNetwork-TTW` model consistently outperformed a simpler ensemble model and the seasonal average model across all weeks of the 2017/2018 season (Fig F in S1 Text).

The `FSNetwork-TTW` model showed a higher performance in both training and testing phase than the CDC baseline ensemble model, `FluSight-unweighted_avg`. This multi-model ensemble contained forecasts from 28 models submitted to the FluSight competition in 2017/2018. While some of these 28 models submitted to the CDC were or contained versions of the 21 models in our performance-based FluSight Network multi-model ensemble, over two-thirds of the models submitted to the CDC were not represented in the FluSight Network components. In 2017/2018, the `FSNetwork-TTW` model earned an average forecast score of 0.337 while the `FluSight-unweighted_avg` model earned an average forecast score of 0.321 (Fig 4).

In the 2017/2018 season, the top models from each contributing research team showed considerable variation in performance across the different prediction targets and regions (Fig 5). However, the `FSNetwork-TTW` model showed lower variability in performance than other methods. Across all 77 pairs of targets and regions, the `FSNetwork-TTW` model was the only

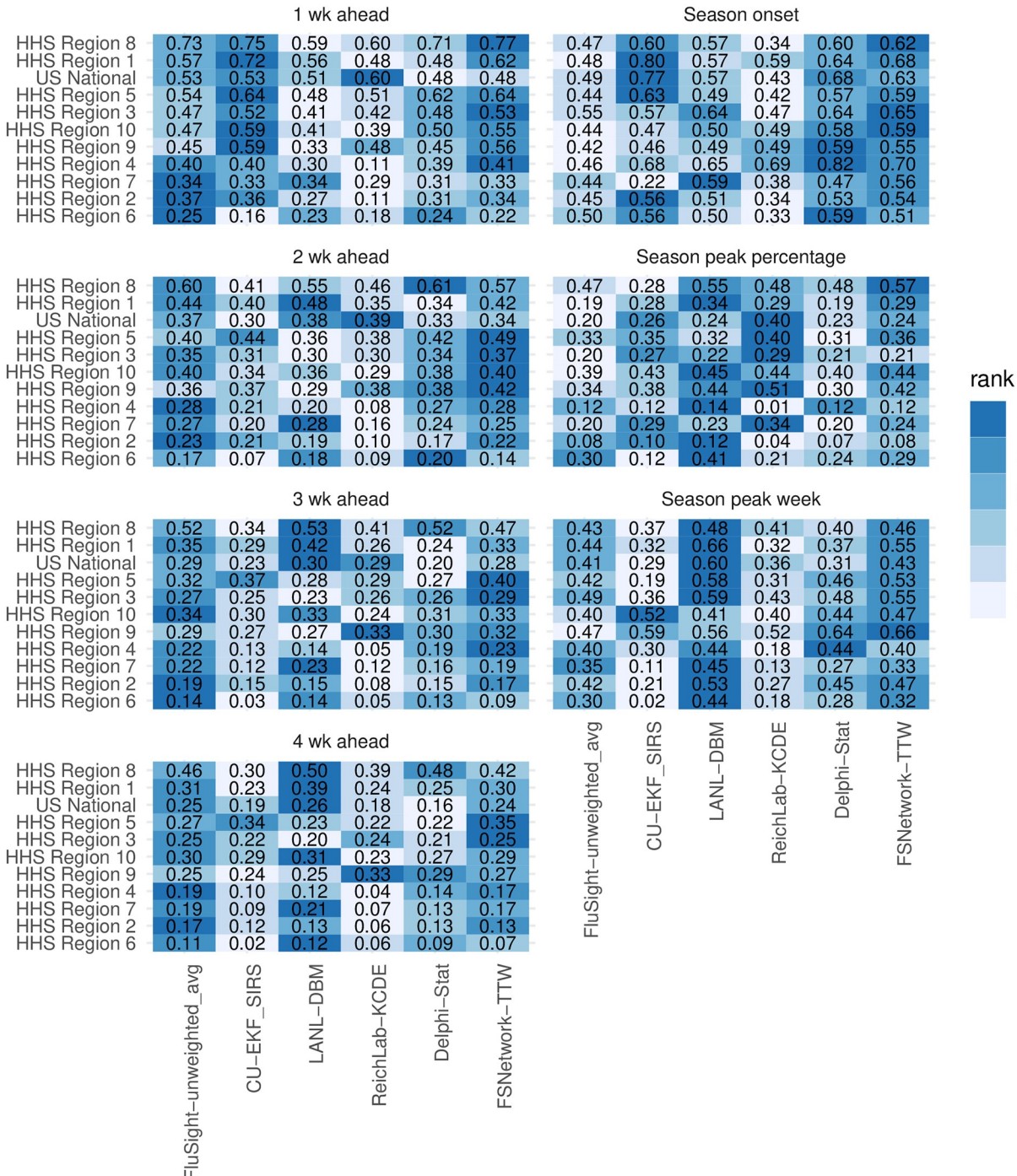

**Fig 5. Average forecast scores and ranks by target and region for 2017/2018.** Models shown include the `FSNetwork-TTW` model, the top performing model from each team during the training phase and the unweighted average of all FluSight models received by CDC. Color indicates model rank in the 2017/2018 season (darker colors indicates higher rank and more accurate forecasts) and the forecast score (rounded to two decimal places) is printed in each cell. Regions are sorted with the most predictable region overall (i.e. highest forecast scores) at the top.

one of the six selected models that never had the lowest forecast score. Additionally, it only had the second lowest score twice. While our ensemble model did not always have the best score in each target-region pair, its consistency and low variability across all combinations secured it the top average score.

Despite being optimized for high forecast score values, the `FSNetwork-TTW` showed robust performance in the 2017/2018 season across other performance metrics that measure forecast calibration and accuracy. Overall, the `FSNetwork-TTW` model ranked second among selected models in both RMSE and average bias, behind the `LANL-DBM` model (Fig A in S1 Text). For example, during the scoring period of interest across all regions in the 2017/2018 season, the `FSNetwork-TTW` model's point estimates for season onset were on average less than half a week after the true value (average bias = 0.38 week) and for 1-week ahead ILI the estimates were underestimated by less than one-quarter of a percentage point (average bias = -0.23 ILI%).

According to the probability integral transform metric [29, 30], the `FSNetwork-TTW` model was well-calibrated for all four week-ahead targets (Fig B in S1 Text). It was slightly less well-calibrated for peak performance, and showed indications of having too narrow predictive distributions over the 2017/2018 season. Over the entire training period prior to the 2017/2018 season, the `FSNetwork-TTW` model calibration results suggested that in general the model was a bit conservative, with often a too wide predictive distribution (Fig C in S1 Text).

A new ensemble using the same components but taking into account their performance in the 2017/2018 season would have different weights. Components that received substantial weight in the original ensemble but did particularly poorly in the 2017/2018 season saw the largest drop in weight (Fig D in S1 Text). Overall, three components were added to the list of six existing components that received more than 0.001 weight for seasonal targets: `CU-EAKFC_SEIRS`, `CU-EKF_SEIRS`, and `ReichLab-SARIMA2`. One component (`ReichLab-SARIMA2`) was added to the list of eight existing components that received more than 0.001 weight for week-ahead targets.

### Ensemble accuracy for peak forecasts

Forecast accuracy around the time of peak incidence is an important indicator of how useful a given model can be in real-time for public health decision-makers. To this end, we evaluated the scores of the `FSNetwork-TTW` ensemble model in each region during the 13 weeks centered around the peak week (Fig 6). Forecast scores of the peak percentage 6, 5, and 4 weeks before the peak week were lower than in past seasons, assigning on average 0.05, 0.06, and 0.05 probability to the eventually observed value, respectively. However, at and after the peak week this probability was over 0.70, quite a bit higher than average accuracy in past seasons.

Similarly, for peak week the average forecast scores improved as the peak week approached. With the exception of a large dip in accuracy in HHS Region 7 just after the peak occured (due to revisions to observed wILI data in the weeks surrounding peak), the forecast scores for peak week tended to be high in the weeks following peak. The average score in more than half of the regions was greater than 0.75 for all weeks after peak.

### Discussion

Multi-model ensembles hold promise for giving decision makers the ability to use "one answer" that combines the strengths of many different modeling approaches while mitigating their weaknesses. This work presents the first attempt to systematically combine infectious disease forecasts from multiple research groups in real-time using an approach that factors in the past performance of each component method. Of the 29 models submitted to the CDC in 2017/2018 as part of their annual FluSight forecasting challenge, this ensemble was the second-highest scoring model overall. The top scoring model was an ensemble of human judgment forecasts [31].

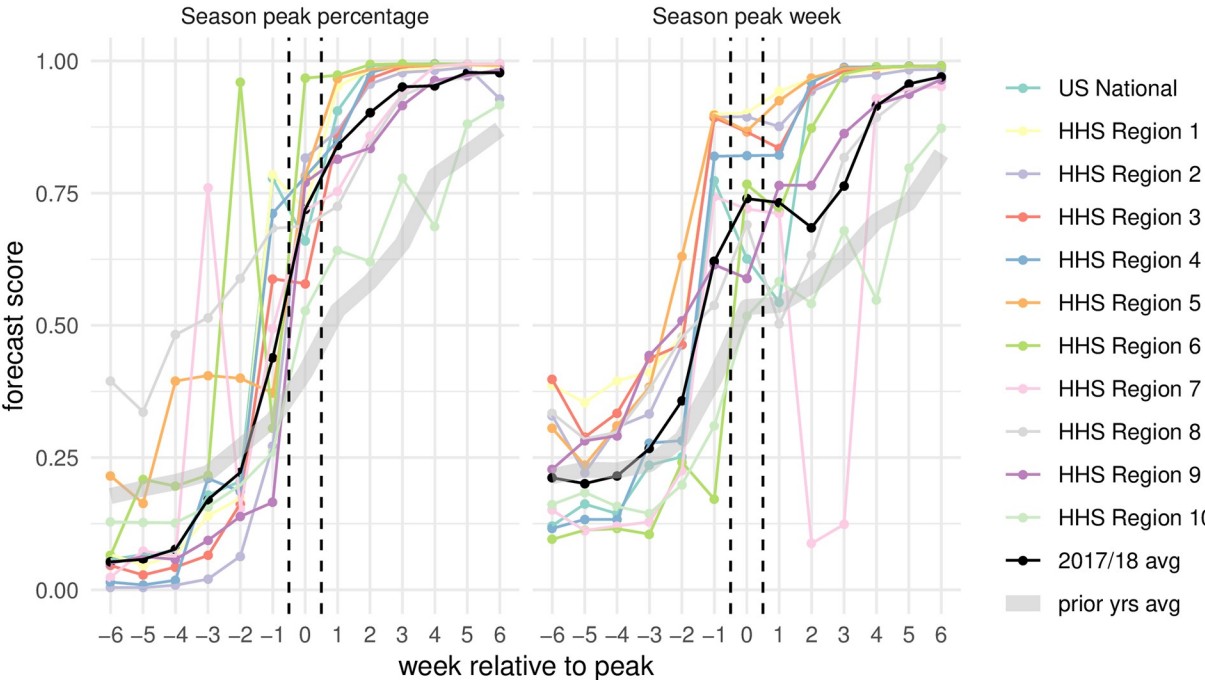

**Fig 6. Forecast score for the `FSNetwork-TTW` model in 2017/2018 by week relative to peak.** Scores for the two peak targets in each region were aligned to summarize performance relative to the peak week. On the x-axis, zero indicates the peak week and positive values represent weeks after the peak week. The black line indicates the overall geometric average across all regions. The grey band represents the geometric average across all regions and all seasons prior to 2017/2018.

By working across disciplines and research groups and by incorporating experts from government, academia and industry, this collaborative effort showed success in bringing measurable improvements in forecast accuracy and reductions in variability. We therefore are moving substantially closer to forecasts that can and should be used to complement routine, ongoing public health surveillance of infectious diseases. In the 2018/2019 influenza season, based on results from this study, the CDC used forecasts from the FluSight Network ensemble model in internal and external communication and planning reports.

Even in a very unusual influenza season, the multi-model ensemble approach presented here outperformed all components overall and did not see a large reduction in overall performance compared to performance during the training seasons. This bodes well for the long-term robustness of models such as this one, compared to single components that show higher variability in performance across specific years, regions, and targets. During the training and test phases, the weighted ensemble approaches outperformed two equal weight ensembles: one constructed based on FluSight Network models presented here (Table A in S1 Text) and one constructed by the CDC using a wider array of models [26]. This clearly illustrates the value of incorporating information on prior performance of component models when constructing a multi-model ensemble.

As shown by the FluSight Network Target-Type Weights component weighting structure presented above (Fig 3), no one model was ever used by the multi-model ensemble as the best answer and the ensemble instead relied on a combination of components to optimize performance. The ensemble assigned non-negligible weight to 11 of the 21 models available and the updated ensemble weights including the 2017/2018 performance would have added one model to that list. This work highlights the importance of incorporating different models into a single

forecast based on past performance. Moving forward, it will be vital to develop and sustain a robust ecosystem of forecasting models for infectious disease that represent many different methods, data sources, and model structures. In particular, we see opportunities in developing incorporating pathogen-specific data into models [32] and using more spatially structured and multi-scale approaches [11, 33]. The inclusion of mutiple forecasting approaches and use of past performance in determining an ensemble structure reduces the risk of relying on a single probabilistic forecast and therefore strengthens the case for incorporating forecasts into real-time decision-making.

While the multi-model ensemble approach described here works well for seasonal pathogens with multiple seasons of retrospective data available, it would be more limited in an emerging pandemic scenario. In these settings, there may not be any historical data on how models have performed nor reliable real-time data to train on. However, adaptive weighting approaches that dynamically update the weights over the course of a season or epidemic could remove the requirement that all models have a substantial track-record of performance. Preliminary work on adaptive weighting has shown some promise, though such approaches still rely on accurately reported real-time data [34]. Furthermore, a simple average of forecasts remains available in such situations and, as illustrated by the relatively strong performance of the FluSight Network Equal Weights model and the CDC's unweighted FluSight ensemble, can still offer advantages over individual models.

One risk of complex ensemble approaches is that they may be "overfit" to the data, resulting in models that place too much emphasis on one approach in a particular scenario or setting. This is a particular concern in applications such as this one, where the number of observations is fairly limited (hundreds to thousands of observations instead of hundreds of thousands). Against this backdrop, the relative simplicity of the FluSight Network Target-Type weights model is a strength, as there is less danger of these models being overfit to the data. Additionally, approaches that use regularization or penalization to reduce the number of effective parameters estimated by a particular model have been shown to have some practical utility in similar settings and may also have a role to play in future ensembles for infectious disease forecasting [23].

Formally measuring the quality of forecasts is challenging and the choice of metric can impact how models are constructed. Following the FluSight Challenge guidelines, we used a probabilistic measure of forecast accuracy, the modified log score, as our primary tool for evaluating forecast accuracy. We also assessed point estimate accuracy as a secondary measure (Fig A in S1 Text). It has been shown that the modified log score (i.e. multiple bins considered accurate) used by the CDC is not strictly proper and could incentivize forecasting teams to modify forecast outputs if their goal was only to achieve the highest score possible [35, 36]. Forecasts in the FluSight Network were not modified in such a way [37]. Most component forecasts were optimized for the proper log-score (i.e. single bins considered accurate) while the FluSight Network ensembles were optimized to the modified log score. By using single bin scoring rules to evaluate forecasts, practitioners could ensure that all forecasts were optimized with the same goal in mind. In the case of the FluSight Network forecasts, the CDC has prioritized accuracy in a probabilistic sense over point-estimate accuracy.

Even though we have shown the value in building collaborations between research teams to develop ensemble forecasts, these efforts largely rely on bespoke technological solutions. This challenge is not specific to infectious disease forecasting, but covers many areas of quantitative science. For this project, we built a highly customized solution that relied on GitHub, Travis Continuous Integration server, unix scripts, and model code in R, python, and MatLab. In all, the seven seasons of training data consisted of about 95MB and over 1.5m rows of data per model and about 2GB of forecast data for all models combined. The real-time forecasts for the

2017/2018 season added about 300MB of data. The framework developed by CDC directly facilitated this work by identifying targets, establishing common formats, and establishing a space for this collaboration. To build on this success and move ensemble infectious disease forecasting into a more generalizable, operational phase, technological advancements are necessary to both standardize data sources, model structures, and forecast formats as well as develop modeling tools that can facilitate the development and implementation of component and ensemble models.

With the promise of new, real-time data sources and continued methodological innovation for both component models and multi-model ensemble approaches, there is good reason to believe that infectious disease forecasting will continue to mature and improve in upcoming years. As modeling efforts become more commonplace in the support of public health decision-making worldwide, it will be critical to develop infrastructure so that multiple models can more easily be compared and combined. This will facilitate reproducibility and archiving of single-model forecasts, the creation of multi-model ensemble forecasts, and the communication of the forecasts and their uncertainty to decision-makers and the general public. Efforts such as this, that emphasize real-time testing and evaluation of forecasting models and facilitate close collaboration between public health officials and modeling researchers, are critical to improving our understanding of how best to use forecasts to improve public health response to seasonal and emerging epidemic threats.

## Materials and methods

### Influenza data

Forecasting targets for the CDC FluSight challenge are based on the U.S. Outpatient Influenza-like Illness Surveillance Network (ILINet). ILINet is a syndromic surveillance system that publishes the weekly percentage of outpatient visits due to influenza-like illness, weighted based on state populations (wILI) from a network of more than 2,800 providers. Estimates of wILI are reported weekly by the CDC's Influenza Division for the United States as a whole as well as for each of the 10 Health and Human Services (HHS) regions. Reporting of 'current' wILI is typically delayed by approximately one to two weeks from the calendar date of a doctor's office visit as data are collected and processed, and each weekly publication can also include revisions to prior reported values if new data become available. Larger revisions have been shown to be associated with decreased forecast accuracy [27]. For the US and each HHS Region, CDC publishes an annual baseline level of ILI activity based on off-season ILI levels [2].

### Forecast targets and structure

As the goal was to submit our ensemble forecast in real-time to the CDC FluSight forecasting challenge, we adhered to guidelines and formats set forth by the challenge in determining forecast format. A season typically consists of forecast files generated weekly for 33 weeks, starting with epidemic week 43 (EW43) of one calendar year and ending with EW18 of the following year. Every week in a year is classified into an "MMWR week" (ranging from 1 to 52 or 53, depending on the year) using a standard definition established by the National Notifiable Diseases Surveillance System [38–40]. Forecasts for the CDC FluSight challenge consist of seven targets: three seasonal targets and four short-term or 'week-ahead' targets. The seasonal targets consist of season onset (defined as the first MMWR week where wILI is at or above baseline and remains above it for three consecutive weeks), season peak week (defined as the MMWR week of maximum wILI), and season peak percentage (defined as the maximum wILI value for the season). The short-term targets consist of forecasts for wILI values 1, 2, 3, and 4 weeks

ahead of the most recently published data. With the two-week reporting delay in the publication of ILINet, these short-term forecasts are for the level of wILI occurring 1 week prior to the week the forecast is made, the current week, and the two weeks after the forecast is made. Forecasts are created for all targets for the US as a whole and for each of the 10 HHS Regions (Fig 1A–1C).

For all targets, forecasts consist of probability distributions over bins of possible values for the target. For season onset and peak week, forecast bins consist of individual weeks within the influenza season, with an additional bin for onset week corresponding to a forecast of no onset. For short-term targets and peak intensity, forecast bins consist of levels of observed wILI rounded to the nearest 0.1% (the level of resolution for ILINet publicly reported by the CDC) up to 13%. Formally, the bins are defined as [0.00, 0.05], [0.05, 0.15], . . ., [12.85, 12.95], [12.95, 100].

The CDC has developed a structured format for weekly influenza forecasts. All forecasts for this project used those data standards for all forecasts and this facilitated collaboration among the teams.

## Forecast evaluation

Submitted forecasts were evaluated using the modified log score used by the CDC in their forecasting challenge, which provides a simultaneous measure of forecast accuracy and precision. The log score for a probabalistic forecast $m$ is defined as $\log f_m(z^*|\mathbf{x})$, where $f_m(z|\mathbf{x})$ is the predicted density function from model $m$ for some target $Z$, conditional on some data $\mathbf{x}$ and $z^*$ is the observed value of the target $Z$.

While a proper log score only evaluates the probability assigned to the exact observed value $z^*$, the CDC uses a modified log score that classifies additional values as "accurate". For predictions of season onset and peak week, probabilities assigned to the week before and after the observed week are included as correct, so the modified log score becomes $\log \int_{z^*-1}^{z^*+1} f_m(z|\mathbf{x})dz$. For season peak percentage and the short-term forecasts, probabilities assigned to wILI values within 0.5 units of the observed values are included as correct, so the modified log score becomes $\log \int_{z^*-.5}^{z^*+.5} f_m(z|\mathbf{x})dz$. In practice, and following CDC scoring convention, we truncate modified log scores to be no lower than -10. We refer to these modified log scores as simply log scores hereafter.

Individual log scores can be averaged across different combinations of forecast regions, target, weeks, or seasons. Each model $m$ has an associated predictive density for each combination of region ($r$), target ($t$), season ($s$), and week ($w$). Each of these densities has an accompanying scalar log score, which could be represented as $\log f_{m,r,t,s,w}(z^*_{r,t,s,w}|\mathbf{x})$. These individual log scores can be averaged across combinations of regions, targets, seasons, and weeks to compare model performance.

Following FluSight challenge convention, to focus model evaluation on periods of time that are more relevant for public health decision-making, only certain weeks were included when calculating the average log scores for each target. Forecasts of season onset were included for each region up to six weeks after the observed onset week within that region. Forecasts of peak week and peak intensity were included for all weeks in a region-season until the week in which the wILI measure dropped below the regional baseline level for the final time. Week-ahead forecasts for each region-season were included starting four weeks prior to the onset week through three weeks after the wILI goes below the regional baseline for the final time. All weeks were included for region-seasons that did not have high enough incidence to define a season onset week.

To enhance interpretability, we report exponentiated average log scores which are the geometric mean of probability a model assigned to the value(s) eventually deemed to be accurate. In this manuscript, we refer to these as "average forecast scores". As an example, the average forecast score for model $m$ in season $s$ (as shown in Fig 2), is computed as

$$S_{m,\cdot,\cdot,s,\cdot} = \exp\left(\frac{1}{N}\sum_{r,t,w}\log f_{m,r,t,s,w}(z^*_{r,t,s,w}|\mathbf{x})\right) = \left(\prod_{r,t,w} f_{m,r,t,s,w}(z^*_{r,t,s,w}|\mathbf{x})\right)^{1/N}. \qquad (1)$$

As other forecasting efforts have used mean square error (MSE) or root mean square error (RMSE) of point predictions as an evaluation method, we additionally evaluated the prospective forecasts received during the 2017-2018 season using RMSE. The submitted point forecast was used to score each component, and a point forecast was generated for each FSNetwork model by taking the median of the predicted distribution. For each model $m$, we calculated $RMSE_{m,t}$ for target $t$, averaging over all weeks $w$ in the $s$ = 2017/2018 season and all regions $r$, as $RMSE_{m,t} = \sqrt{\frac{\sum_{r,w}(\hat{z}_{m,r,t,w}-z^*_{r,t,w})^2}{N}}$, where $\hat{z}_{m,r,t,w}$ is the point prediction of model $m$ for observed value $z^*_{r,t,w}$. Average bias is calculated as $bias_{m,t} = \frac{\sum_{r,w}(\hat{z}_{m,r,t,w}-z^*_{r,t,w})}{N}$.

## Ensemble components

To provide training data for the ensemble, four teams submitted between 1 and 9 components each, for a total of 21 ensemble components. Teams submitted out-of-sample forecasts for the 2010/2011 through 2016/2017 influenza seasons. These models and their performance are evaluated in separate work [27]. Teams constructed their forecasts in a prospective fashion, using only data that were available at the time of the forecast. For some data sources (e.g., wILI prior to the 2014/2015 influenza season), data as they were published at the time were not available. In such cases, teams were still allowed to use those data sources while making best efforts to only use data available at the time forecasts would have been made.

For each influenza season, teams submitted weekly forecasts from epidemic week 40 (EW40) of the first year through EW20 of the following year, using standard CDC definitions for epidemic week [38–40]. If a season contained EW53, forecasts were submitted for that week as well. In total, teams submitted 233 individual forecast files representing forecasts across the seven influenza seasons. Once submitted, the forecast files were not updated except in four instances where explicit programming bugs had resulted in numerical issues in the forecast. Teams were explicitly discouraged from re-tuning or adjusting their models for different prior seasons to avoid issues with over-fitting.

Teams utilized a variety of methods and modeling approaches in the construction of their component model submissions (Table A in S1 Text). Seven of the components used a compartmental structure (i.e. Susceptible-Infectious-Recovered) to model the disease transmission process, while other components used more statistical approaches to directly model the observed wILI curve. Six of the components explicitly incorporated additional data sources beyond previous wILI data, including weather data and Google search data. Two components were constructed to represent a seasonal baseline based on historical data only.

Additionally, we obtained the predictive distributions from the CDC-created "unweighted average" model. This ensemble combined all forecast models received by the CDC in real-time in the 2015/2016 (14 models), 2016/2017 (28 models), and 2017/2018 (28 models) seasons [26]. These included models that are not part of the collaborative ensemble effort described in this manuscript, although some variations on the components presented here were also

submitted to the CDC. Including this model allowed us to compare our ensemble accuracy to the model used by the CDC in real-time during these three seasons.

It is important to distinguish ensemble components from standalone forecasting models. Standalone models are optimized to be as accurate as possible on their own by, among other things, using proper smoothing. Ensemble components might be designed to be accurate on their own, or else they may be included merely to complement weak spots in other components, i.e. to reduce the ensemble's variance. Because we had sufficient cross-validation data to estimate ensemble weights for several dozen components, some groups contributed non-smoothed "complementing" components for that purpose (Table A in S1 Text). Such components may perform poorly on their own, yet their contribution to overall ensemble accuracy may still be significant.

It should be noted that ensemble weights are not a measure of ensemble components' standalone accuracy nor do they measure the overall contribution of a particular model to the ensemble accuracy. For example, consider a setting where a component that is identical (or highly similar) to an existing ensemble component with weight $\pi^*$ is added to a given ensemble. The accuracy of the original ensemble can be maintained in a number of ways, including (a) assigning each copy a weight of $\pi^*/2$, or (b) assigning the first copy a weight of $\pi^*$ and the second copy a weight of 0. In both of these weightings, at least one high accuracy ensemble component would be assigned significantly lower weight due to the presence of another identical or similar component. In fact, we saw this in our results since the `Delphi-Stat` model was the top-performing component model but was a linear combination of other Delphi models. It received zero weight in all of our ensemble specifications. Additionally, inclusion of components with small weights can have a large impact on an ensemble's forecast accuracy.

## Ensemble nomenclature

There are several different ways that the term ensemble has been used in practice. In this paper, we use the phrases 'multi-model ensemble' or 'ensemble model' interchangably to refer to models that represent mixtures of separate component models. However, a clear taxonomy of ensemble modeling might distinguish three distinct tiers of ensemble models. First, single-model ensemble methodologies can be used to fit models and make predictions. Examples of these approaches include the component models from Columbia University that use, e.g. Ensemble Average Kalman Filtering, to take weighted averages of model realizations to form predictive distributions (Table A in S1 Text). Second, multi-model ensembles combine component models through techniques such as model stacking (see Methods). Among the models described in this work, one component model (`Delphi-Stat`) is a multi-model ensemble and all of the FluSight Network models are also multi-model ensembles (Table A in S1 Text). Third, the term superensemble has been used for models that combine components that are themselves ensembles (either multi-model or single-model) [16, 41]. Since not all of the components in our approach are ensembles themselves, we chose the term multi-model ensemble to refer to our approach.

## Ensemble construction

All ensemble models were built using a method that combines component predictive distributions or densities using weighted averages. In the literature, this approach has been called stacking [13] or weighted density ensembles [23], and is similar to methods used in Bayesian model averaging [18]. Let $f_c(z_{t,r,w})$ represent the predictive density of ensemble component $c$ for the value of the target $Z_{t,r,w}$, where $t$ indexes the particular target, $r$ indexes the region, and

$w$ indexes the week. We combine these components together into a multi-model ensemble with predictive density $f(z_{t,r,w})$ as follows:

$$f(z_{t,r,w}) = \sum_{c=1}^{C} \pi_{c,t,r} f_c(z_{t,r,w}) \tag{2}$$

where $\pi_{c,t,r}$ is the weight assigned to component $c$ for predictions of target $t$ in region $r$. We require $\sum_{c=1}^{C} \pi_{c,t,r} = 1$ and thereby ensure that $f(z_{t,r,w})$ remains a valid probability distribution.

A total of five ensemble weighting schemes were considered, with varying complexity and number of estimated weights (Table A in S1 Text).

- Equal Weight (`FSNetwork-EW`): This model consisted of assigning all components the same weight regardless of performance and is equivalent to an equally weighted probability density mixture of the components: $\pi_{c,t,r} = 1/C$.

- Constant Weight model (`FSNetwork-CW`): The weights vary across components but have the same value for all targets and regions, for a total of 21 weights: $\pi_{c,t,r} = \pi_c$. For purposes of statistical estimation, we say that the degrees of freedom (df) is $(21 - 1) = 20$. For each set of weights, once 20 weights are estimated the 21st is determined since they must add up to 1.

- Target Type Weight model (`FSNetwork-TTW`): Weights are estimated separately for our two target-types ($tt$), short-term and seasonal targets, with no variation across regions. This results in a total of 42 weights (df = 40): $\pi_{c,t,r} = \pi_{c,tt}$.

- Target Weight model (`FSNetwork-TW`): The weights are estimated separately for each of the seven targets for each component with no variation across regions, resulting in 147 weights (df = 140): $\pi_{c,t,r} = \pi_{c,t}$.

- Target-Region Weight model (`FSNetwork-TRW`): The most complex model considered, this model estiamted weights separately for each component-target-region combination, resulting in 1617 unique weights (df = 1540): $\pi_{c,t,r} = \pi_{c,t,r}$.

Weights were estimated using the EM algorithm (Section 6 in S1 Text) [34]. Weights for components were trained using a leave-one-season-out cross-validation approach on component forecasts from the 2010/2011 through 2016/2017 seasons. Given the limited number of seasons available for cross-validation, we used component model forecast scores from all other seasons as training data to estimate weights for a given test season, even if the training season occured chronologically after the test season of interest.

## Ensemble evaluation

Based on the results of the cross-validation study, we selected one ensemble model as the official FluSight Network entry to the CDC's 2017/2018 influenza forecasting challenge. The criteria for this choice were pre-specified in September of 2017, prior to conducting the cross-validation experiments [28]. Component weights for the `FSNetwork-TTW` model were estimated using all seven seasons of component model forecasts. In real-time over the course of the 2017/2018 influenza season, participating teams submitted weekly forecasts from each component, which were combined using the estimated weights into the FluSight Network model and submitted to the CDC. The component weights for the submitted model remained unchanged throughout the course of the season.

## Reproducibility and data availability

To maximize the reproducibility and data availability for this project, the data and code for the entire project are publicly available. The project is available on GitHub [42], with a permanent repository stored on Zenodo [43]. Code for specific models are either publicly available or available upon request from the modeling teams, with more model-specific details available at the related citations (Table A in S1 Text). Retrospective and real-time forecasts from the FluSight Network may be interactively browsed on the website http://flusightnetwork.io. Additionally, this manuscript was dynamically generated using R version 3.6.0 (2019-04-26), Sweave, knitr, and make. These tools enable the intermingling of manuscript text with R code that run the central analyses, automatically regenerate parts of the analysis that have changed, and minimize the chance for errors in transcribing or translating results [44, 45].

## Supporting information

**S1 Text. Supplementary methods and results.**
(PDF)

## Acknowledgments

The findings and conclusions in this report are those of the authors and do not necessarily represent the official position of the Centers for Disease Control and Prevention, Defense Advanced Research Projects Agency, Defense Threat Reduction Agency, the National Institutes of Health, National Institute for General Medical Sciences, National Science Foundation, or Uptake Technologies.

## Author Contributions

**Conceptualization:** Nicholas G. Reich, Teresa K. Yamana, Evan L. Ray, Dave Osthus, Sasikiran Kandula, Logan C. Brooks, Matthew Biggerstaff, Michael A. Johansson, Roni Rosenfeld, Jeffrey Shaman.

**Data curation:** Nicholas G. Reich, Craig J. McGowan, Abhinav Tushar, Evan L. Ray, Dave Osthus, Sasikiran Kandula, Logan C. Brooks.

**Formal analysis:** Nicholas G. Reich, Craig J. McGowan, Teresa K. Yamana, Evan L. Ray, Dave Osthus, Logan C. Brooks.

**Funding acquisition:** Nicholas G. Reich, Roni Rosenfeld, Jeffrey Shaman.

**Investigation:** Nicholas G. Reich, Craig J. McGowan, Teresa K. Yamana.

**Methodology:** Nicholas G. Reich, Logan C. Brooks, Roni Rosenfeld.

**Project administration:** Nicholas G. Reich.

**Software:** Nicholas G. Reich, Abhinav Tushar, Logan C. Brooks.

**Supervision:** Nicholas G. Reich, Roni Rosenfeld, Jeffrey Shaman.

**Validation:** Nicholas G. Reich, Craig J. McGowan, Teresa K. Yamana, Abhinav Tushar, Evan L. Ray, Dave Osthus, Sasikiran Kandula, Logan C. Brooks, Willow Crawford-Crudell, Graham Casey Gibson, Evan Moore, Rebecca Silva.

**Visualization:** Nicholas G. Reich, Craig J. McGowan, Evan Moore.

**Writing – original draft:** Nicholas G. Reich, Craig J. McGowan.

**Writing – review & editing:** Nicholas G. Reich, Craig J. McGowan, Teresa K. Yamana, Abhinav Tushar, Evan L. Ray, Dave Osthus, Sasikiran Kandula, Logan C. Brooks, Willow Crawford-Crudell, Graham Casey Gibson, Evan Moore, Rebecca Silva, Matthew Biggerstaff, Michael A. Johansson, Roni Rosenfeld, Jeffrey Shaman.

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
