## [Decision Letter · Decision Letter 0]

21 Aug 2019

Dear Dr Reich,

Thank you very much for submitting your manuscript, 'Accuracy of real-time multi-model ensemble forecasts for seasonal influenza in the U.S.', to PLOS Computational Biology. As with all papers submitted to the journal, yours was fully evaluated by the PLOS Computational Biology editorial team, and in this case, by independent peer reviewers. The reviewers appreciated the attention to an important topic but identified some aspects of the manuscript that should be improved.

We would therefore like to ask you to modify the manuscript according to the review recommendations before we can consider your manuscript for acceptance. Your revisions should address the specific points made by each reviewer and we encourage you to respond to particular issues Please note while forming your response, if your article is accepted, you may have the opportunity to make the peer review history publicly available. The record will include editor decision letters (with reviews) and your responses to reviewer comments. If eligible, we will contact you to opt in or out.raised.

- Supporting Information uploaded as separate files, titled 'Dataset', 'Figure', 'Table', 'Text', 'Protocol', 'Audio', or 'Video'.

We hope to receive your revised manuscript within the next 30 days. If you anticipate any delay in its return, we ask that you let us know the expected resubmission date by email at ploscompbiol@plos.org.

Sincerely,

Virginia E. Pitzer, Sc.D.

Deputy Editor

PLOS Computational Biology

Rob De Boer

Deputy Editor

PLOS Computational Biology

[LINK]

Reviewer's Responses to Questions

**Comments to the Authors:**

Reviewer #1: Reich and colleagues describe the development and validation of a multi-model ensemble approach for flu forecasting. This work has arisen out of a collaborative network of groups participating in the CDC’s flu forecasting challenge. It is hard to find much to fault with this study. The paper is clearly written, and the analysis plan and execution are rigorous and well-thought out. This study provides a gold-standard for how forecasting work should be performed, with clear, pre-specified outcomes, extensive development with cross validation, and an out-of-sample, real-time test of the method. I have only minor comments to help with the interpretation of some of the results.

I understand that the authors were constrained in the metrics that were reported based on the criteria for the CDC’s contest. However, in some instances, it would be easier to interpret if the results were presented differently. For instance, the authors demonstrate that the TTW model performs better than the equal weight model. But it is difficult to evaluate from the forecast scores whether these improvements are meaningful. Expressing some of the results in terms of the units of measurement might be helpful. For instance, the estimate of peak week was off by an average of X weeks for TTW compared with Y weeks for equal-weighting.

Is there a way to visualize/summarize how the forecast accuracy for the different models changes throughout the season as data accrue for other forecasting targets (similar to what is done in figure 6 for peak week)?

Given that the ensembles performed worse than the prior-season average for predicting peak intensity early in the season (Fig 6), would there be any benefit to including the prior years average itself in the ensemble?

It seems that the ensemble weights here do not vary through the season and are based only on the cross-validation period. Would there be any benefit to using the cross-validation weights as a starting point and then allowing the ensemble weights to vary each week as the data accrue?

Could the authors comment on the criteria chosen by CDC for evaluating accuracy and precision as well as the forecasting targets and whether there are modifications that they would suggest based on their experience?

Reviewer #2: In this article, the authors demonstrate that a multi-model ensemble forecast was able to provide accurate forecasts of influenza-like illness activity over the course of the 2017/18 United States influenza season. They describe how a range of multi-model ensembles were created and trained on past years' data, how the best-performing ensemble was identified, and then entered into the CDC FluSight challenge. This ensemble not only out-performed each of the individual models in the ensemble, but also came second overall in the challenge, despite the 2017/18 season being highly unusual. These kinds of collaborative efforts and methodological developments are critical if such forecasts are to become a part of routine public health surveillance.

Comments:

1. The motivation for selecting only one ensemble model as the official FluSight Network entry in the 2017/18 challenge is perfectly reasonable and pragmatic.

But it would be great to see even a cursory comparison of the selected ensemble model (FSNetwork-TTW) and the four other ensemble models (FSNetwork-EW, FSNetwork-CW, FSNetwork-TW, FSNetwork-TRW) for 2017/18, even though those other models weren't officially entered into the competition. In particular, it would be really interesting to see whether the performance similarities between these models in the training phase were also evident in such an unusual influenza season, or if the differences between these models conveyed any (dis)advantages in this scenario.

2. In the result section (page 8, lines 179-182) the authors state:

"Overall, the FSNetwork-TTW model ranked second among selected models in both RMSE and average bias, behind the LANL-DBM model (see Appendix), suggesting that using separate weighting schemes for point estimates and predictive distribution may be valuable."

This gave me a moment's pause. The reasoning is explained in more detail in section 2.1 of the supplementary material, and some of this detail could be included in the main text. Perhaps just a reminder that ensemble weights were chosen to maximize log scores, rather than point-estimate errors, would be sufficient.

3. In figure 4, cells with dark blue background could have values shown in white text or a light color, rather than black text, to make it easier to read. This also applies to Figure 1 in the supplementary material.

4. In the methods section (page 15, lines 388-389) a reference is missing:

"Second, multi-model ensembles combine component models through techniques such as model stacking (see Section )."

**Have all data underlying the figures and results presented in the manuscript been provided?**

Reviewer #1: Yes

Reviewer #2: Yes

PLOS authors have the option to publish the peer review history of their article (what does this mean?). If published, this will include your full peer review and any attached files.

Reviewer #1: No

Reviewer #2: No

---

## [Decision Letter · Decision Letter 1]

14 Oct 2019

Dear Dr Reich,

We are pleased to inform you that your manuscript 'Accuracy of real-time multi-model ensemble forecasts for seasonal influenza in the U.S.' has been provisionally accepted for publication in PLOS Computational Biology.

In the meantime, please log into Editorial Manager at https://www.editorialmanager.com/pcompbiol/, click the "Update My Information" link at the top of the page, and update your user information to ensure an efficient production and billing process.

One of the goals of PLOS is to make science accessible to educators and the public. PLOS staff issue occasional press releases and make early versions of PLOS Computational Biology articles available to science writers and journalists. PLOS staff also collaborate with Communication and Public Information Offices and would be happy to work with the relevant people at your institution or funding agency. If your institution or funding agency is interested in promoting your findings, please ask them to coordinate their releases with PLOS (contact ploscompbiol@plos.org).

Thank you again for supporting Open Access publishing. We look forward to publishing your paper in PLOS Computational Biology.

Sincerely,

Virginia E. Pitzer, Sc.D.

Deputy Editor

PLOS Computational Biology

Rob De Boer

Deputy Editor

PLOS Computational Biology

Reviewer's Responses to Questions

**Comments to the Authors:**

Reviewer #1: The authors have thoroughly responded to my comments.

Reviewer #2: The authors have thoroughly addressed all of my original comments, and I have no further comments for them to address. This is a very well designed study, reported in a very clear manuscript.

**Have all data underlying the figures and results presented in the manuscript been provided?**

Reviewer #1: Yes

Reviewer #2: Yes

PLOS authors have the option to publish the peer review history of their article (what does this mean?). If published, this will include your full peer review and any attached files.

Reviewer #1: No

Reviewer #2: No

---

## [Editor Report · Acceptance letter]

12 Nov 2019

PCOMPBIOL-D-19-01044R1 

Accuracy of real-time multi-model ensemble forecasts for seasonal influenza in the U.S.

Dear Dr Reich,

I am pleased to inform you that your manuscript has been formally accepted for publication in PLOS Computational Biology. Your manuscript is now with our production department and you will be notified of the publication date in due course.

With kind regards,

Matt Lyles
